# Differential Associated Factors for Inadequate Receipt of Components and Non-Use of Antenatal Care Services among Adolescent, Young, and Older Women in Nigeria

**DOI:** 10.3390/ijerph20054092

**Published:** 2023-02-24

**Authors:** Esther Awazzi Envuladu, Abukari Ibrahim Issaka, Mansi Vijaybhai Dhami, Biniyam Sahiledengle, Kingsley Emwinyore Agho

**Affiliations:** 1Department of Community Medicine, College of Health Sciences, University of Jos, Jos 930003, Nigeria; 2School of Health Sciences, Western Sydney University, Locked Bag 1797, Penrith, NSW 2751, Australia; 3The Children’s Hospital at Westmead, Locked Bag 4001, Westmead, NSW 2145, Australia; 4Translational Health Research Institute (THRI), Western Sydney University, Penrith, NSW 2571, Australia; 5Department of Public Health, Madda Walabu University Goba Referral Hospital, Bale-Goba 4540, Ethiopia; 6African Vision Research Institute, Westville Campus, University of KwaZulu-Natal, Durban 3629, South Africa

**Keywords:** antenatal care, inadequate, adolescent, women, Nigeria

## Abstract

Nigerian women continue to die in childbirth due to inadequate health services such as antenatal care (ANC). Among other factors, the inadequate receipt or non-use of ANC appears to be associated with the age of women, remoteness, and poor households. This cross-sectional study aimed to compare the factors associated with inadequate receipt of the components and non-use of ANC among pregnant adolescents, and young and older women in Nigeria. Data for this study were from the 2018 Nigeria Demographic and Health Survey (NDHS) and covered a weighted total of 21,911 eligible women. Survey multinomial logistic regression analyses that adjusted for cluster, and survey weights were conducted to examine factors associated with adolescent, young, and older women. Adolescent women reported a higher prevalence of inadequate receipts and non-use of ANC than young and older women. Increased odds of inadequate receipt of the components of ANC were associated with residence in the North–East region and rural areas for all three categories of women. For adolescent women, the increased odds of inadequate receipt of the components of ANC were associated with delivering a baby at home and a big problem with distance to health facilities. Limited education or no schooling was associated with the increased odds of receiving inadequate ANC among older women. Implementing interventions to improve maternal and child health care should focus on the factors associated with the increased odds of receipt of inadequate or non-use of ANC services among Nigerian adolescent women, particularly those living in rural areas in the North–East region.

## 1. Introduction

Adequate antenatal care (ANC) services remain one of the four pillars of the safe motherhood initiative, aimed at providing care that will improve the survival of women and babies [1,2,3]. The World Health Organization (WHO) recently recommended at least eight ANC visits for all pregnant women, irrespective of age and risk status, to ensure a more positive pregnancy outcome [2].

Antenatal care is a strategy geared towards curtailing the high maternal and newborn mortality, which Nigeria records at an unacceptably high rate of 39 deaths per 1000 live births in the 2018 Nigeria Demographic and Health Survey (NDHS), contributing to the global burden of maternal and newborn death [4]. There is evidence of a direct link between maternal health and perinatal and neonatal outcomes. Integrating maternal and newborn care into routine health services in low- and middle-income countries, including Nigeria, may improve overall health outcomes in women and children [5]. Providing accessible quality ANC has been identified as a major intervention to minimize maternal, perinatal, and neonatal mortality [6,7]. A significant component of regular ANC is the maternal and fetal assessment in identifying high-risk pregnancies, which require referral to the next level of maternal care [2,7]. Worldwide, almost 86% of pregnant women receive at least one ANC care from a skilled health professional. Nonetheless, only 65% received the recommended eight ANC visits [8]. Nigeria has consistently had a low ANC coverage rate. Recent data showed a slight increase of about 3% from the previous 58% in 2008 NDHS to 61% in 2013 NDHS, but this is far below the recommended 90% coverage rate required for a meaningful reduction in newborn and maternal death through early detection and treatment of some causes of death [9].

The documented national average ANC coverage rate in Nigeria is not the exact reflection of the country’s status considering the variations across regions, states, and urban and rural locations [10]. Additionally, there are variations in the coverage of ANC between adolescent pregnant women and older women. Pregnancy and childbearing are critical for adolescent women and, therefore, of great public health concern. Adolescents are at greater odds of adverse pregnancy and delivery complications than older women [10], and they are about twice as likely to die from maternal-related causes than non-adolescents [11,12].

Every year, about 16 million girls aged 15–19 give birth, 95% are in low and middle-income countries, particularly sub-Saharan Africa, with 2 million girls giving birth before the age of 15 in these countries [11]. Adolescent pregnancy is still high in Nigeria. According to the most recent Nigerian Demographic Health Survey (NDHS), the birth rate among adolescents is 106 births per 1000 women and even higher in some regions, like the northern regions, requiring intense ANC among adolescents [4].

Adolescents are less likely to receive four or more ANC visits and have a skilled healthcare provider at ANC clinics than women in the older age groups [13]. This is because they face multiple barriers that reduce their access to maternal healthcare information and services [13,14], Adolescent women are less likely to utilize maternal healthcare services such as ANC, which is a complex phenomenon influenced by many factors. These factors are the lack of adolescent-centered services and socioeconomic challenges found to be more prevalent in adolescents, thereby lowering their access to maternal health services [15,16].

To meet the sustainable development goals (SDGs 3) target of improving maternal health and the reduction of maternal mortality to a reasonable level by 2030, it is important to understand the actual access to ANC services by women, including adolescents, who have been found to be one of the vulnerable groups, in terms of motherhood. Furthermore, utilization of ANC has been found to be associated with the age of women; the risk of its inadequacy is higher among older women (aged 31 years or older) [17,18], although a past study from Vietnam observed that age did not affect the utilization of ANC [19].

A number of studies examined the utilization of ANC services and factors associated with them in Nigeria [20,21,22]. One study compared factors associated with the underutilization of ANC services in urban and rural Nigeria [22]. However, there has yet to be a recent study in Nigeria comparing factors associated with inadequate and non-utilization of ANC services among women of different age groups and emphasizing adolescents. This research, therefore, aimed to undertake a comparative study on the factors associated with inadequate and non-utilization of ANC services among women aged 15–19 years, 20–34 years, and 35–49 years. Findings from our study may provide vital information to policymakers and healthcare administrators to design and/or revise strategies geared towards providing adequate ANC services to Nigerian women, particularly adolescents.

## 2. Materials and Methods

### 2.1. Data Sources

The analysis is a secondary data analysis of the 2018 Nigeria Demographic and Health Survey (NDHS-6). The survey was conducted by National Population Commission (NPC) with technical assistance provided by the Inner-City Fund (ICF) International. The representative sample consisted of women aged 15–49 years [4]. Two-stage sampling was used to retrieve data for both the rural and urban areas. The first stage consisted of designating the village units as primary sample units and the urban units as census enumeration areas for sampling the households. Across Nigeria, nearly 42,000 households were selected for sampling, and the response rate was 99% [4].

Our analysis was restricted to the women with the most recent singleton live-birth infants within the five years preceding the survey. A weighted total of 21,911 women were considered for the survey. Additionally, our restriction of the sample aimed to reduce the recall bias of women regarding their previous pregnancies.

### 2.2. Outcome Variables

The World Health Organization (WHO) recommends a number of appropriate components of ANC to be patronized by all pregnant women at ANC clinics. These are blood pressure measurement; screening for conditions and diseases such as anaemia, STIs (particularly syphilis), and HIV infection; tetanus vaccination; intermittent preventive treatment for malaria during pregnancy (IPTp); deworming treatment; and iron–folic-acid supplements [23]. Furthermore, the WHO has recommended that all pregnant women attend an appropriate number of ANC visits. The recommendation is broken down into the following categories: (a) ≥8 ANC visits (b) 4–7 ANC visits, and (c) <4 ANC visits from a skilled health provider for the most recent birth [24].

In our current study, six of the components were used: (a) being given iron tablets/syrup, (b) being given drugs for intestinal parasites, (c) vaccination against tetanus toxoid, (d) checking blood pressure, (e) having blood samples taken, and (f) had urine samples were taken. The current study considered two outcome variables: (i) non-ANC use (no ANC visit and none of the recommended WHO components) and (ii) receipt of inadequate ANC (receiving at most eight ANC visits and receiving less than six recommended WHO components). In our analysis, adequate receipt of ANC was assigned “1”, of inadequate receipt ANC was categorized as “2” and non-use of ANC was assigned “3” and were examined by three categories: adolescent women (15–19 years), young women (20–34 years), and older women (35–49 years).

### 2.3. Potential Confounding Variables

In this current study, we chose the potential confounding factors based on a behavioral model for health services utilization, proposed by Andersen, and grouped the factors potentially associated with ANC [25]. The Andersen model has been broadly used to examine health service utilization factors [1,18,26,27]. Furthermore, the choice of potential confounding factors was informed by the available variables in the datasets of the NDHS-6. The independent variables were categorized into external, predisposing (socio-demographic and health knowledge), and enabling factors [1,28]. The underlying conceptual framework for this current study is presented in Figure 1.

### 2.4. Statistical Analysis

We conducted our statistical analyses using STATA/MP version 17 (Stata Corp, College Station, TX, USA) [29]. The “Svy” commands were employed to allow for adjustments for the cluster-sampling design and weight. We conducted frequency tabulations to describe the data used in the study and the distributions of inadequate ANC. We used the Taylor series linearization method in the surveys to estimate confidence intervals (CIs) around prevalence estimates of receipt of inadequate quality ANC.

We performed a four-stage model as part of the multinomial logistic regression analysis to calculate the adjusted odds ratios of receipt of inadequate ANC. We entered the basic factors in the first model and undertook a manually executed elimination procedure to examine factors associated with the receipt of inadequate ANC at a 0.05 significance level. In the second model, we added the significant factors in the first stage to the underlying factors, followed by the elimination process. The third and fourth stages used a similar approach for the other factors.

Any co-linearity in the final model was tested and reported. Furthermore, we computed the odds ratios with 95% confidence intervals to assess the factors associated with the independent (possible confounding) variables.

## 3. Results

### 3.1. Characteristics of the Sample among Adolescent, Young, and Older Women

Table 1 presents the characteristics of the participants among adolescent, young, and older women. According to Table 1, most women had no formal education and were not working, which was mostly among adolescent women (60.5% and 59.4%). The majority of women across all the age brackets were married, with 89.7%, 94.3%, and 94.8% in the 15–19 years, 20–34 years and 35–49 years age brackets, respectively. Most adolescent women delivered at home 72.1% against 41.4% and 43.1% younger and older women. Most women in all three age brackets (84.8% for 15–19 years, 88.5% for 20–34 years, and 87.9% for 35–49 years) had no problem seeking permission to attend a medical facility on their own, but more reported getting money to pay for health services as a big problem and this was more among the age group 15–19 years (52.3%, 47.7%, 48.6%). The majority of the women aged 15–19 years (adolescent women) lived in rural areas (79.9%), and poor households (62.3%). Most adolescent women did not have access to the mass media (94.6%, 62.4%, and 72.5% of them did not have access to newspapers/magazines, radio, and television, respectively). Only 6.5% of adolescent women attended the minimum of eight ANC clinics, as against 15.6% and 17.2% for younger and older women.

### 3.2. Components of ANC Services Received by Participants

Table 2 presents the receipt of ANC services by pregnant Nigerian women. Overall, more than 6 out of every 10 (63.9%) participants received iron tablets or syrup. The lowest prevalence of iron tablets or syrup intake was among the women in the 15–19 years age bracket (62.6%). Less than 20% of the general participants (16.7%) took any intestinal parasite drugs. The lowest prevalence of participants who took drugs for intestinal parasites was the adolescent (15–19 years) group (12.7%). Nearly 70% of the general participants (69.6%) had tetanus injections before delivery, with the lowest prevalence (60.7%) among the adolescent group. Most participants had their blood pressure checked during pregnancy (71.1%). The adolescent participants had the lowest prevalence of blood pressure checks (60.6%). More than two-thirds of the general participants had their blood samples taken during pregnancy; The adolescent participants had the lowest prevalence of women who had their blood samples taken (53.7%). A little over 65% (65.3%) of the general participants had their urine samples taken during pregnancy, and the adolescent participants had the lowest prevalence of those who had their urine samples taken (53.9%).

### 3.3. Prevalence and 95% Confidence Intervals (Cis) of Receipt of ANC Services

Figure 2 shows the prevalence and 95% confidence intervals of receipt of ANC services among adolescent, young, and older women. The prevalence of non-use of ANC ranged from 30.1% to 36.5%, with adolescent women reporting the highest prevalence and young women with the lowest prevalence. However, this prevalence among women did not differ statistically because the confidence intervals overlap. Over 50% of all women reported inadequate receipt of ANC.

### 3.4. Factors Associated with Inadequate Receipt and Non-Use of ANC Services

#### 3.4.1. Women Aged 15–19 Years

Among the women aged 15–19 years, increased odds of inadequate receipt of ANC were associated with delivery at home [(adjusted odds ratios (OR): 3.40; 95% confidence interval (CI): 1.52, 7.61)], acquisition of medical assistance for any distance to a health facility [OR: 2.06; 95% CI: (0.91, 4.69)], living in the North–East region [OR: 5.02; 95% CI: (1.38, 18.26)], and residence in a rural area [OR: 3.40; 95% CI: (1.52, 7.61)] (Table 3). Decreased odds of inadequate receipt of ANC were associated with residence in the South–South region [OR: 0.09; 95% CI: (0.03, 0.36)] among women aged 15–19 years.

Increased odds of non-use of ANC were associated with delivery at home [OR: 4.70; 95% CI: (2.08, 11.35)], distance to a health facility being a big problem [OR: 16.53; 95% CI: (6.71, 40.73)] and living in a rural area [OR: 16.53; 95% CI: (6.71, 40.73)]. Residence in the South–South region was associated with decreased odds of non-use of ANC among women aged 15–19 years [OR: 0.09; 95% CI: (0.03, 0.36)].

#### 3.4.2. Women Aged 20–34 Years

Among women aged between 20 and 34 years, increased odds of inadequate receipt of ANC were significantly associated with no schooling [OR: 1.52; 95% CI: (1.24, 1.87)], non-use of contraceptives [OR: 1.44; 95% CI: (1.21, 1.71)], non-intention for pregnancy [OR: 1.44; 95% CI: (1.21, 1.71)], belonging to middle-class households [OR: 1.30; 95% CI: (1.05, 1.61)], residing in the North–East region [OR: 4.28; 95% CI: (2.58, 7.08)], and living in a rural area [OR: 1.57; 95% CI: (1.31, 1.88)] (Table 4). Decreased odds of inadequate receipt of ANC were significantly associated with not reading newspapers [OR: 0.70; 95% CI: (0.51, 0.96)] and living in the South–South region [OR: 0.12; 95% CI: (0.09, 0.18)] among women aged 20–34 years (Table 4).

Increased odds of non-use of ANC were associated with no schooling [OR: 3.09; 95% CI: (2.34, 4.08)], non-intention for pregnancy [OR: 2.37; 95% CI: (1.41, 3.99)], non-use of contraceptives [OR: 1.73; 95% CI: (1.42, 2.11)], not listening to the radio [OR: 1.43; 95% CI: (1.14, 1.78)], getting permission to attend a health facility being a big problem [OR: 1.62; 95% CI: (1.17, 2.24)], getting the money needed for treatment being a big problem [OR: 1.31; 95% CI: (1.03, 1.66)], being from a poor household [OR: 1.51; 95% CI: (1.10, 2.08)], and living in a rural area [OR: 4.06; 95% CI: (3.33, 4.95)].

#### 3.4.3. Women Aged 35–49 Years

Among women aged 35–49 years, increased odds of inadequate receipt of ANC were associated with no schooling [OR: 1.52; 95% CI: (1.00, 2.31)], living in the North–East region [OR: 3.77; 95% CI: (2.02, 7.03)], being from a poor household [OR: 1.87; 95% CI: (1.29, 2.70)], and living in a rural area [OR: 2.15; 95% CI: (1.59, 2.90)] (Table 5). Decreased odds of inadequate receipt of ANC were associated with being employed [OR: 0.54; 95% CI: (0.38, 0.75)] among women aged 35–49 years (Table 5).

Increased odds of non-use of ANC among women aged 35–49 years were associated with having no schooling [OR: 3.07; 95% CI: (2.00, 4.70)], distance to a health facility being a big problem [OR: 1.81; 95% CI: (1.31, 2.51)], non-use of contraceptives [OR: 1.35; 95% CI: (1.05, 1.75)], living in the North–West region [OR: 2.22; 95% CI: (1.30, 3.80)], being from a poor household [OR: 1.75; 95% CI: (1.16, 2.65)], and living in a rural area [OR: 5.61; 95% CI: (3.99, 7.89)].

## 4. Discussion

This current study aimed to compare the factors associated with receipt of inadequate or non-use of ANC services among adolescent, younger, and older women in Nigeria, using data extracted from the NDHS-6. The lowest prevalence of receipt of adequate utilization of ANC services and the highest prevalence of receipt of both inadequate and non-use of ANC services was among the adolescent participants. Furthermore, the majority of the participants who lived in rural areas were adolescents. Rural residence was a common predictor of receipt of inadequate or non-use of ANC among all three age brackets of participants. Additionally, more education was needed to be provided as a common predictor of receipt of inadequate or non-use of ANC services among the older participants (20–34 years and 35–49 years).

Previous studies revealed that women with lower educational levels usually have limited knowledge about ANC and more difficulties accessing ANC services [1,30,31,32]. This was confirmed in our current study, where participants’ limited education level was associated with increased odds of receiving inadequate or non-use of ANC services among older women (both women in the 20–34 years and 35–49 years brackets). The association between the level of education and the receipt of inadequate ANC services is consistent with findings from past studies in developing countries [33,34]. The impact of education on the use of maternal health care services could be explained in several ways. Educated women are more likely to know the long-term benefits of utilizing skilled maternity care services for both mother and baby than their less-educated counterparts. The woman’s education is likely to enhance female autonomy, which may consequently help develop greater confidence and capability to make appropriate decisions about their health [35]. There is also the likelihood of educated women seeking higher quality health services and having a higher ability to use health care inputs that provide adequate care [36].

This study found that among women of all three age brackets, residence in a rural area was associated with increased odds of inadequate receipt and non-use of ANC services, consistent with a past study from Bangladesh [37]. The reasons for the decreased odds of receipt of inadequate or non-use of ANC services among urban women compared to their rural counterparts are the ease of access to services and increased knowledge of the adverse health outcomes due to pregnancy-related complications. Moreover, the tendency of rural women to be influenced by traditional practices which contravene modern health care is higher than that of their urban counterparts [38].

In this current study, delivering a baby at home was associated with increased odds of receiving no or inadequate quality ANC among adolescent participants. This is consistent with findings from a past study in Timor-Leste [39], which revealed that home delivery was positively associated with an increased risk of receipt of inadequate and non-use of ANC among adolescent women. There are cultural practices and beliefs associated with the choice of a place of delivery [40]; if a woman’s previous birth was at home and there were no complications, there is the tendency for her not to attend subsequent ANC visits, or she may limit the frequency of such visits. Women who deliver their babies at home are those with higher parity who feel they are experienced in delivering a baby; and tend not to seek assistance, especially when there are no complications [41]. Such women consider child delivery a natural procedure, so they are likely to feel confident about delivering their babies at home. Furthermore, there is evidence that the association between home delivery and non-ANC use may be attributed to a woman’s bad experience with health workers during her previous deliveries [42]. The decision of women to give birth at home may be due to the distance from their homes to health facilities or their socioeconomic status [43]. For example, a study conducted in rural Tanzania observed that women were significantly more likely to deliver at home if the nearest health facility to their home was outside their village [44].

In our current study, with regards to the quality of ANC services received, the prevalence of all six variables of quality ANC services, namely, administering of iron tablets/syrup, drugs for intestinal parasites, and tetanus injection before delivery, as well as blood pressure testing and supplying of both blood and urine samples, was lowest among the adolescent participants.

The extant literature reveals that the increased health risk of a mother and her baby is associated with adolescent pregnancy and childbearing [45]. For instance, a past study reported higher rates of preterm deliveries, prolonged labor, and cephalic pelvic disproportion among teenage girls compared to older women [46]. Other past studies also observed that teenage women’s odds of death during pregnancy and childbirth are higher [47,48]. Apart from the mother, adolescent pregnancy also poses a danger to the baby. Babies born to adolescent women have an increased risk of low birth weight [48]. Babies born to adolescent women in low-income countries also face a 50% chance of stillbirth or death before they turn one month, compared to women aged above 20 years [49]. From these observations, it may be suggested that while minimizing the rate of teenage pregnancies is crucial, a concurrent goal would be to support teenage women towards a safe and healthy pregnancy, delivery, and baby through increased access and utilization of skilled maternal health care [50].

As with participants from the older age brackets, the increased odds of receipt of inadequate or non-use of ANC services were associated with residence in the North–East region of Nigeria. This ought to be expected because of the wide gap in socioeconomic development between the northern and southern parts of the country [51]. There is also evidence to the effect that the North–East region was one of the regions with relatively low ANC services [52].

A major strength of this study was the utilization of NDHS-6 data, population-based and nationally representative, with a large sample size. However, the study’s cross-sectional nature limits the ability to draw any causal inferences, and we could not rule out any residual confounding resulting from other socio-demographic and psychosocial-related determinants. Furthermore, most of the information inquired about in the study was self-reported, which could have led to the under-reporting of certain determinants. Additionally, data were collected retrospectively from participants, which may have resulted in recall bias.

## 5. Conclusions

Findings from this current study highlight the receipt of inadequate utilization of ANC services among women of all ages, particularly adolescents in Nigeria. The situation appeared to be worse in rural areas, where most of the women live, and most are poor. Results of the study reveal that the factors influencing the utilization of ANC services are mostly socioeconomic- and demographic-related. The rural–urban and regional variations in the ANC services revoke the need to make quality maternal health service facilities available in Nigeria. Consequently, government and other stakeholders should conduct mass media campaigns to highlight the benefits of utilizing ANC services and harmful traditional practices. This can be achieved by informing, educating, and communicating with the rural, lower-educated, and the poor in a bid to promote safe motherhood and a happy and healthy reproductive life among adolescents in Nigeria.

## Figures and Tables

**Figure 1 ijerph-20-04092-f001:**
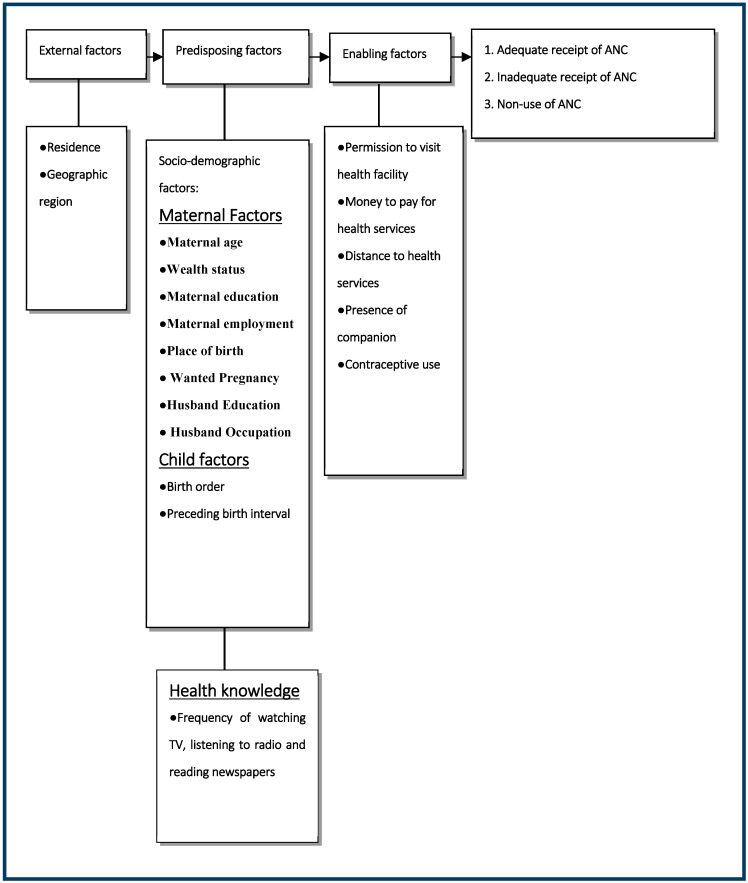
Conceptual framework of factors associated with no ANC or receipt of inadequate ANC services in Nigeria.

**Figure 2 ijerph-20-04092-f002:**
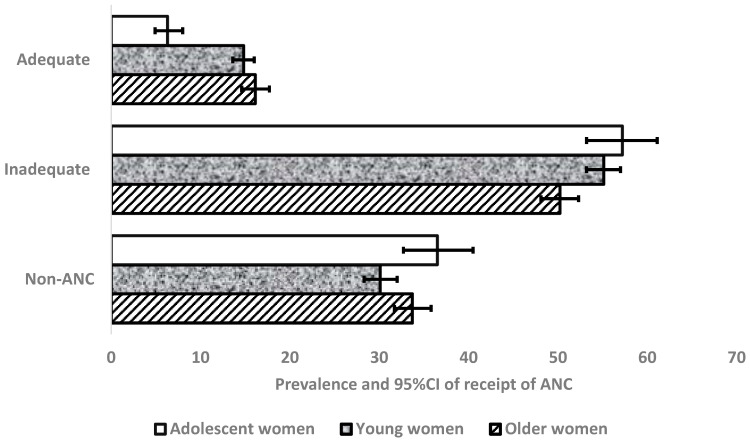
Prevalence and 95% confidence level of receipts of ANC by adolescent, young, and older women.

**Table 1 ijerph-20-04092-t001:** Distribution of characteristics among adolescent, young, and older women in Nigeria (*n* = 21,872), 2018 NDHS.

Characteristics	Adolescent (N * = 1210)	Young (N * = 14,649)	Older (N * = 6013)
	(*n*, %) *	(*n*, %) *	(*n*, %) *
External Factors			
Geographical region			
North Central	143 (11.9)	2119 (14.5)	763 (12.7)
North East	288 (23.8)	2595 (17.7)	975 (16.2)
North West	581 (48.0)	5058 (34.5)	2000 (33.3)
South East	69 (5.7)	1387 (9.5)	675 (11.2)
South West	77 (6.3)	1361 (9.3)	574 (9.6)
South-South	51 (4.2)	2129 (14.5)	1025 (17.1)
Type of place of residence			
Urban	243 (20.1)	5836 (39.8)	2611 (43.4)
Rural	967 (79.9)	8813 (60.2)	3402 (56.6)
Predisposing factors			
Maternal Factors			
Household Wealth Index			
Rich	42 (3.4)	2523 (17.2)	1217 (20.2)
Middle	414 (34.3)	5845 (39.9)	2275 (37.8)
Poor	754 (62.3)	6282 (42.9)	2521 (41.9)
Mother’s working status			
Non-working	718 (59.4)	5347 (36.5)	1450 (24.1)
Working	492 (40.7)	9302 (63.5)	4563 (75.9)
Husband’s Education			
Secondary	359 (33.8)	7149 (52.6)	2606 (46.4)
Primary	139 (13.1)	1747 (12.9)	940 (16.7)
No education	564 (53.2)	4697 (34.6)	2072 (36.9)
Husband’s occupation			
Non-agricultural work	566 (46.8)	8480 (57.9)	3297 (54.8)
Agricultural work	466 (38.5)	4820 (32.9)	2167 (36.1)
Unemployed	178 (14.7)	1349 (9.2)	549 (9.1)
Mother’s Education			
Secondary	321 (26.6)	6476 (44.2)	2058 (34.2)
Primary	157 (13.0)	2043 (13.9)	1083 (18.0)
No education	731 (60.5)	6131 (41.9)	2872 (47.8)
Mother’s marital status			
Currently married	1085 (89.7)	13,816 (94.3)	5701 (94.8)
Never in marriage	125 (10.3)	833 (5.7)	312 (5.2)
Place of delivery			
Home	872 (72.1)	6067 (41.4)	2590 (43.1)
Health Facility	338 (27.9)	8582 (58.6)	3423 (56.9)
Wanted pregnancy at the time			
Then	1046 (86.5)	13,000 (88.7)	5172 (86.0)
Later	158 (13.1)	1431 (9.8)	341 (5.7)
No more	6 (0.5)	218 (1.5)	500 (8.3)
Child Factors			
Preceding birth interval			
None	968 (80.0)	2696 (18.4)	128 (2.1)
Short/long interval	242 (20.0)	11,954 (81.6)	5886 (97.9)
Birth Order			
1	964 (79.7)	2666 (18.2)	123 (2.1)
2 to 4	246 (20.3)	8387 (57.3)	1445 (24.0)
5		3597 (24.6)	4446 (73.9)
Health knowledge			
Frequency of reading newspapers or magazine			
At least once a week	17 (1.4)	468 (3.2)	220 (3.7)
Less than once a week	48 (4.0)	1272 (8.7)	522 (8.7)
Never	1144 (94.6)	12,909 (88.1)	5272 (87.7)
Frequency of listening to the radio			
At least once a week	222 (18.4)	4130 (28.2)	1937 (32.2)
Less than once a week	233 (19.2)	3697 (25.2)	1406 (23.4)
Never	755 (62.4)	6822 (46.6)	2669 (44.4)
Frequency of watching television			
At least once a week	149 (12.3)	4168 (28.5)	1640 (27.3)
Less than once a week	184 (15.3)	2486 (17.0)	1125 (18.7)
Never	877 (72.5)	7996 (54.6)	3248 (54.0)
Enabling Factors			
Seek permission to visit health services			
Big problem	184 (15.2)	1692 (11.6)	726 (12.1)
Not a big problem	1026 (84.8)	12,958 (88.5)	5287 (87.9)
Getting money to pay for health services			
Big problem	632 (52.3)	6990 (47.7)	2925 (48.6)
Not a big problem	578 (47.8)	7659 (52.3)	3088 (51.4)
Distance to a health facility			
Big problem	425 (35.1)	4022 (27.5)	1667 (27.7)
Not a big problem	785 (64.9)	10,628 (72.6)	4346 (72.3)
Accompany to a health facility			
Big problem	263 (21.7)	2292 (15.7)	875 (14.6)
Not a big problem	947 (78.3)	12,357 (84.4)	5138 (85.5)
Antenatal care services			
Antenatal Clinical visits			
8+	76 (6.5)	2165 (15.6)	969 (17.2)
4 to 7	415 (35.4)	5547 (39.9)	2118 (37.5)
1 to 3	279 (23.8)	2564 (18.4)	911 (16.1)
None	402 (34.3)	3629 (26.1)	1652 (29.2)
Uptake of receipt of Antenatal care			
Receipt of adequate	76 (6.3)	2165 (14.8)	969 (16.1)
Receipt of inadequate	692 (57.2)	8069 (55.1)	3017 (50.2)
Non-ANC	442 (36.5)	4416 (30.1)	2026 (33.7)

N * = Weighted total count; *n* * = weighted count and % * = weighted percent.

**Table 2 ijerph-20-04092-t002:** Components of antenatal care services received by adolescent, young, and older women during the last pregnancy, 2018 NDHS.

Variable	Adolescent (N * = 1210)		Young (N * = 14,649)		Older (N * = 6013)		All Women (N * = 21,872)	
	*n* *	% * (95% CI)	*n* *	% * (95% CI)	*n* *	% * (95% CI)	*n* *	% * (95% CI)
Given iron tablets/syrup								
Yes	758	62.6 (58.6, 66.5)	10,336	70.6 (68.6, 72.4)	4067	67.6 (65.4, 69.8)	15,161	69.3 (67.5, 71.1)
Taken drugs for intestinal parasites								
Yes	153	12.7 (10.6, 15.1)	2575	17.6 (16.5, 18.7)	932	15.5 (14.3, 16.8)	3660	16.7 (15.8, 17.7)
Had tetanus injections before birth								
Yes	734	60.7 (56.7, 64.5)	10,423	71.2 (69.3, 72.9)	4079	67.8 (65.7, 69.9)	15,235	69.6 (67.9, 71.4)
Blood pressure taken								
Yes	733	60.6 (56.6, 64.5)	10,579	72.2 (70.2, 74.2)	4222	70.2 (68.1, 72.3)	15,535	71.0 (69.1, 72.9)
Had blood sample taken								
Yes	650	53.7 (49.9, 57.4)	9894	67.5 (65.5, 69.5)	3949	65.7 (63.4, 67.9)	14,493	66.3 (64.3, 68.2)
Had urine sample taken								
Yes	652	53.9 (49.9, 57.9)	9744	66.5 (64.4, 68.5)	3890	64.7 (62.5, 66.9)	14,286	65.3 (63.4, 67.2)

N * = Weighted total count; *n* * = weighted count; and % * = weighted percent.

**Table 3 ijerph-20-04092-t003:** Factors associated with receipt of inadequate receipt of ANC and non-use of ANC among Adolescent women group (15–19 years), 2018 NDHS.

Characteristics	Inadequate Receipt of ANC				Non-ANC			
	Unadjusted OR	*p*-Value	Adjusted OR	*p*-Value	Unadjusted OR	*p*-Value	Adjusted OR	*p*-Value
Place of delivery								
Health Facility	1.00		1.00		1.00		1.00	
Home	7.78 (4.12, 14.68)	<0.001	3.40 (1.52, 7.61)	<0.001	31.86 (14.94, 67.97)	<0.001	4.70 (2.08, 11.35)	<0.001
Distance to a health facility								
Not a big problem	1.00		1.00		1.00		1.00	
Big problem	2.26 (1.15, 4.45)	0.018	2.06 (0.91, 4.69)	0.083	5.51 (2.80, 10.88)	<0.001	16.53 (6.71, 40.73)	<0.001
Geographical region								
North Central	1.00		1.00		1.00		1.00	
North East	7.22 (2.01, 25.93)	0.002	5.02 (1.38, 18.26)	0.014	4.65 (1.24, 17.39)	0.022	2.10 (0.54, 8.10)	0.281
North West	3.43 (1.12, 10.50)	0.031	2.23 (0.67, 7.45)	0.189	2.72 (0.87, 8.47)	0.085	1.23 (0.36, 4.24)	0.733
South East	0.21 (0.08, 0.56)	0.002	0.24 (0.09, 0.66)	0.005	0.10 (0.03, 0.32)	<0.001	0.18 (0.06, 0.59)	0.005
South West	0.29 (0.10, 0.84)	0.023	0.21 (0.08, 0.61)	0.004	0.24 (0.08, 0.67)	0.007	0.13 (0.05, 0.40)	<0.001
South South	0.07 (0.02, 0.20)	<0.001	0.08 (0.03, 0.26)	<0.001	0.05 (0.02, 0.16)	<0.001	0.09 (0.03, 0.36)	0.001
Type of place of residence								
Urban	1.00		1.00		1.00		1.00	
Rural	7.78 (4.12, 14.69)	<0.001	3.40 (1.52, 7.61)	0.003	31.87 (14.94, 67.97)	<0.001	16.53 (6.71, 40.73)	<0.001

**Table 4 ijerph-20-04092-t004:** Factors associated with receipt of inadequate components ANC and non-ANC use among young women aged 20–34 years, 2018 NDHS.

Characteristics	Inadequate Reciept of ANC				Non-ANC			
	Unadjusted OR	*p*-Value	Adjusted OR	*p*-Value	Unadjusted OR	*p*-Value	Adjusted OR	*p*-Value
Maternal Factors								
Mother’s working status								
Not working	1.00		1.00		1.00		1.00	
working	0.50 (0.42, 0.59)	<0.001	0.97 (0.81, 1.16)	0.763	0.39 (0.33, 0.48)	<0.001	0.80 (0.66, 0.97)	0.025
Mother’s Education								
Secondary	1.00		1.00		1.00		1.00	
Primary	3.04 (2.48, 3.75)	<0.001	1.52 (1.24, 1.87)	<0.001	3.39 (2.68, 4.27)	<0.001	1.57 (1.25, 1.96)	<0.001
No education	11.21 (8.89, 14.14)	<0.001	1.35 (1.02, 1.78)	0.034	28.38 (21.95, 36.70)	<0.001	3.09 (2.34, 4.08)	<0.001
Contraceptive use								
Yes	1.00		1.00		1.00		1.00	
No	2.95 (2.53, 3.44)	<0.001	1.44 (1.21, 1.71)	<0.001	4.98 (4.17, 5.95)	<0.001	1.73 (1.42, 2.11)	<0.001
Wanted pregnancy at the time								
Then	1.00		1.00		1.00		1.00	
Later	0.58 (0.48, 0.70)	<0.001	1.05 (0.86, 1.27)	0.657	0.51 (0.42, 0.62)	<0.001	1.07 (0.86, 1.33)	0.559
No more	0.63 (0.40, 0.97)	0.038	1.36 (0.83, 2.22)	0.224	1.12 (0.71, 1.78)	0.626	2.37 (1.41, 3.99)	0.001
Frequency of reading newspapers or magazine								
At least once a week	1.00		1.00		1.00		1.00	
Less than once a week	0.76 (0.55, 1.04)	0.087	0.67 (0.48, 0.94)	0.020	1.12 (0.72, 1.74)	0.606	0.87 (0.53, 1.41)	0.566
Never	1.91 (1.42, 2.56)	<0.001	0.70 (0.51, 0.96)	0.026	4.73 (3.30, 6.79)	<0.001	0.76 (0.50, 1.15)	0.194
Frequency of watching television								
At least once a week	1.00		1.00		1.00		1.00	
Less than once a week	1.14 (0.97, 1.35)	0.107	1.17 (0.98, 1.40)	0.087	1.52 (1.26, 1.83)	<0.001	1.34 (1.09, 1.64)	0.005
Never	2.61 (2.16, 3.14)	<0.001	1.03 (0.84, 1.26)	0.797	5.05 (4.09, 6.25)	<0.001	1.43 (1.14, 1.78)	0.002
Seek permission to visit health services								
Not a big problem	1.00		1.00		1.00		1.00	
Big problem	0.75 (0.56, 1.02)	0.064	0.85 (0.63, 1.15)	0.304	2.25 (1.66, 3.06)	<0.001	1.62 (1.17, 2.24)	0.003
Getting money to pay for health services								
Not a big problem	1.00		1.00		1.00		1.00	
Big problem	1.46 (1.24, 1.72)	<0.001	1.10 (0.89, 1.35)	0.367	2.35 (1.94, 2.85)	<0.001	1.31 (1.03, 1.66)	0.027
Child Factors								
Preceding birth interval								
None	1.00		1.00		1.00		1.00	
Short/long interval	2.05 (1.79, 2.35)	<0.001	1.22 (1.05, 1.41)	0.008	2.58 (2.19, 3.04)	<0.001	1.23 (1.03, 1.48)	0.023
Household Factors								
Geographical region								
North Central	1.00		1.00		1.00		1.00	
North East	5.59 (3.36, 9.31)	<0.001	4.28 (2.58, 7.08)	<0.001	4.11 (2.35, 7.19)	<0.001	1.73 (1.01, 2.96)	0.046
North West	3.97 (2.63, 6.01)	<0.001	2.96 (1.95, 4.48)	<0.001	4.45 (2.79, 7.10)	<0.001	1.77 (1.15, 2.73)	0.010
South East	0.29 (0.20, 0.42)	<0.001	0.39 (0.27, 0.57)	<0.001	0.13 (0.08, 0.20)	<0.001	0.41 (0.27, 0.62)	<0.001
South West	0.18 (0.13, 0.26)	<0.001	0.20 (0.15, 0.28)	<0.001	0.25 (0.17, 0.36)	<0.001	0.42 (0.30, 0.59)	<0.001
South South	0.08 (0.06, 0.12)	<0.001	0.12 (0.09, 0.18)	<0.001	0.13 (0.09, 0.19)	<0.001	0.41 (0.29, 0.59)	<0.001
Household Wealth Index								
Rich	1.00		1.00		1.00		1.00	
Middle	2.82 (2.25, 3.53)	<0.001	1.30 (1.05, 1.61)	0.014	2.76 (2.24, 3.40)	<0.001	1.02 (0.79, 1.31)	0.870
Poor	10.66 (8.11, 14.02)	<0.001	1.23 (0.93, 1.63)	0.154	22.73 (17.2, 30.03)	<0.001	1.51 (1.10, 2.08)	0.011
Type of place of residence								
Urban	1.00		1.00		1.00		1.00	
Rural	4.78 (4.04, 5.65)	<0.001	1.57 (1.31, 1.88)	<0.001	14.24 (11.4, 17.78)	<0.001	4.06 (3.33, 4.95)	<0.001

**Table 5 ijerph-20-04092-t005:** Factors associated with receipt of inadequate components of ANC and non-ANC use among older women aged 35–49 years, 2018 NDHS.

Characteristics	Inadequate Receipt of ANC				Non-ANC			
	Unadjusted OR	*p*-Value	Adjusted OR	*p*-Value	Unadjusted OR	*p*-Value	Adjusted OR	*p*-Value
Mother’s working status								
Not working	1.00		1.00		1.00		1.00	
Working	0.47 (0.36, 0.61)	<0.001	0.88 (0.62, 1.25)	0.472	0.28 (0.21, 0.36)	<0.001	0.54 (0.38, 0.75)	<0.001
Mother’s Education								
Secondary	1.00		1.00		1.00		1.00	
Primary	2.24 (1.68, 2.98)	<0.001	1.14 (0.85, 1.53)	0.388	2.65 (2.04, 3.45)	<0.001	1.18 (0.87, 1.59)	0.284
No education	12.76 (9.26, 17.58)	<0.001	1.52 (1.00, 2.31)	0.049	28.9 (20.75, 40.27)	<0.001	3.07 (2.00, 4.70)	<0.001
Contraceptive use								
Yes	1.00		1.00		1.00		1.00	
No	2.74 (2.26, 3.31)	<0.001	1.22 (0.99, 1.50)	0.058	4.31 (3.49, 5.34)	<0.001	1.35 (1.05, 1.75)	0.019
Distance to the health facility								
Not a big problem	1.00		1.00		1.00		1.00	
Big problem	1.31 (1.00, 1.73)	0.052	1.07 (0.80, 1.44)	0.649	2.68 (1.99, 3.61)	<0.001	1.81 (1.31, 2.51)	<0.001
Geographical region								
North Central	1.00		1.00		1.00		1.00	
North East	6.21 (3.39, 11.35)	<0.001	3.77 (2.02, 7.03)	<0.001	5.31 (2.84, 9.93)	<0.001	1.94 (1.04, 3.63)	0.038
North West	5.56 (3.39, 9.11)	<0.001	3.30 (1.98, 5.53)	<0.001	6.27 (3.68, 10.68)	<0.001	2.22 (1.30, 3.80)	0.003
South East	0.36 (0.25, 0.53)	<0.001	0.51 (0.35, 0.74)	<0.001	0.14 (0.09, 0.22)	<0.001	0.38 (0.24, 0.61)	<0.001
South West	0.25 (0.17, 0.38)	<0.001	0.31 (0.21, 0.46)	<0.001	0.37 (0.24, 0.57)	<0.001	0.72 (0.47, 1.10)	0.131
South South	0.12 (0.08, 0.17)	<0.001	0.17 (0.12, 0.25)	<0.001	0.17 (0.11, 0.26)	<0.001	0.48 (0.31, 0.74)	0.001
Household Wealth Index								
Rich	1.00		1.00		1.00		1.00	
Middle	2.97 (2.30, 3.84)	<0.001	1.41 (1.10, 1.80)	0.007	2.5 (1.79, 3.49)	<0.001	0.92 (0.66, 1.27)	0.597
Poor	12.92 (9.09, 18.36)	<0.001	1.87 (1.29, 2.70)	0.001	22.51 (15.34, 33.01)	<0.001	1.75 (1.16, 2.65)	0.008
Type of place of residence								
Urban	1.00		1.00		1.00		1.00	
Rural	7.78 (4.12, 14.69)	<0.001	2.15 (1.59, 2.90)	<0.001	31.87 (14.94, 67.97)	<0.001	5.61 (3.99, 7.89)	<0.001

## Data Availability

The study was based on an analysis of existing survey datasets that are available to apply online, with all identifier information removed. Written informed consent for the present analysis was not necessary because secondary data analysis did not involve interaction with the participants. The data collection methods for the NDHS-6 data used in this analysis, including the consent process, have been previously described [4]. Written informed consent for the present analysis was not necessary because secondary data analysis did not involve interaction with the participants.

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
