# Peer review of "Differential Associated Factors for Inadequate Receipt of Components and Non-Use of Antenatal Care Services among Adolescent, Young, and Older Women in Nigeria"

_ijerph, 2023, doi:10.3390/ijerph20054092_

Round 1
Reviewer 1 Report
The authors of this manuscript have worked on identifying factors associated with the low uptake of ANC services in Nigeria. This is a well-conducted study, but I have some minor considerations:
1. Line 16: please expand on the a-priori factors affecting service uptake.
2. There are some repeated sentences in the abstract. For example: "Older women reported the lowest prevalence of non-utilisation of ANC than adolescents and young women." and "The highest prevalence of receipt of adequate and inadequate ANC services was higher among adolescent and young women compared with older women."
Minor proofreading errors:
3. Please remove the period just at the beginning of the abstract.
4. Line 28: there's double-spacing before "Implementing". Double-spacing can also be seen in line 48.
5. Line 46: please add space before the citation and check the rest of the document for similar errors.
6. The authors should only spell out the abbreviations one time and then use the abbreviation consistently. Several problems exist with NDHS and LMICs. Please amend accordingly.
7. Line 280-2: there's a format error.
Author Response
ANC services in Nigeria. This is a well-conducted study, but I have some minor considerations:
Query: Line 16: please expand on the a-priori factors affecting service uptake.
Response: I have added two more a-priori factors – see yellow highlighted in the Abstract
Query: There are some repeated sentences in the abstract. For example: "Older women reported the lowest prevalence of non-utilisation of ANC than adolescents and young women." and "The highest prevalence of receipt of adequate and inadequate ANC services was higher among adolescent and young women compared with older women."
Response: Thanks. This has been modified including making the figure much clearer.
Minor proofreading errors:
Query: Please remove the period just at the beginning of the abstract.
Response: Thanks and now edited.
Query: Line 28: there's double-spacing before "Implementing". Double-spacing can also be seen in line 48.
Response: Thanks and now edited.
Query: Line 46: please add space before the citation and check the rest of the document for similar errors.
Response: Thanks and now edited.
Query: The authors should only spell out the abbreviations one time and then use the abbreviation consistently. Several problems exist with NDHS and LMICs. Please amend accordingly.
Response: Thanks and we have added a section to the List of abbreviations used.
Query: Line 280-2: there's a format error.
Response: Thanks and now edited.
Reviewer 2 Report
The work addresses a worrying and necessary scientific area, for which the team is thanked for this study. It addresses gender and health, areas of first level of study.
The research has been approached methodologically adequately, with a good theoretical analysis and quite timely conclusions. Here are some considerations to take into account, if deemed appropriate:
· The introduction mentions the Sustainable Development Goals, but does not specify exactly what it is (indicate with number).
· Perhaps it would be appropriate to state more explicitly the criteria for inclusion and exclusion of the sample
· Figure 1 should be edited with higher quality
· Table 1 and Table 4 are very dense, perhaps you could reduce and make some graph too, so that the data is more readable.
· It is recommended not to put all the data that are already given in the tables or figures, in the text stick only to those most significant data for the study
Author Response
Query: The introduction mentions the Sustainable Development Goals, but does not specify exactly what it is (indicate with number).
Response: Thanks and now edited.
Query: Perhaps it would be appropriate to state more explicitly the criteria for inclusion and exclusion of the sample
Response: We already included exclusion criteria but have now modified – see the line: 109-110:
Our analysis was restricted to the women with the most recent singleton live-birth infants within the five years preceding the survey
Query: Figure 1 should be edited with higher quality
Response: Figure 1 has been edited for clarity and higher quality.
Query: Table 1 and Table 4 are very dense, perhaps you could reduce and make some graph too, so that the data is more readable.
Response: The statistical approach used in this manuscript is the multinomial logistic regression and it is not possible to produce graphs using this type of statistical method.
Query: It is recommended not to put all the data that are already given in the tables or figures, in the text stick only to those most significant data for the study
Response: As I stated above because we used multinomial logistic regression, some factors that is significant for non-use of ANC may not be significant for inadequate receipt of ANC but because we modelled them together, remove one because it is not significant will make it difficult for the readers to follow and in addition, it is also important to produce non-significant, especially for those researchers who liked producing meta-analysis.
Reviewer 3 Report
This retrospective review on ANC utilization in Nigeria is based on secondary analysis of data of 2018 based drawn from a population of about 22000 women. The summary and methodology is acceptable. The statistical analysis and results are well presented . Table and Figures are acceptable in number and comprehensive. In Table 1, 2 , kindly include the N for each of the three groups studied (Column Heading).
The findings from this study shows findings similar to other low and middle income countries. Clearly adolescent pregnant women, health literacy and access to ANC services are factors surfacing as areas to be tackled . Perhaps, in the conclusions, we should suggest remediable actions apart from what is mentioned. It would be good to indicate the availability of ANC by population in rural and urban areas in the study zones, and if there are any reasons why clients do not access these services e.g., remoteness, financial issues, timing of clinics, services rendered.
As this study is based on data from 2018, we need to be cautious in drawing the conclusions as set out. Has the governing body taken any measures to improve health literacy? Have there been a move to build more ANC closer to the residence of clients, have there been any improvements made according the six areas of 'Safe Motherhood Initiatives'? Data based on 2018 is a limitation of this study, this needs to be stated.
Author Response
Query: This retrospective review on ANC utilization in Nigeria is based on secondary analysis of data of 2018 based drawn from a population of about 22000 women. The summary and methodology is acceptable. The statistical analysis and results are well presented . Table and Figures are acceptable in number and comprehensive. In Table 1, 2 , kindly include the N for each of the three groups studied (Column Heading).
Response: Thanks and now edited.
Query: The findings from this study shows findings similar to other low and middle income countries. Clearly adolescent pregnant women, health literacy and access to ANC services are factors surfacing as areas to be tackled . Perhaps, in the conclusions, we should suggest remediable actions apart from what is mentioned. It would be good to indicate the availability of ANC by population in rural and urban areas in the study zones, and if there are any reasons why clients do not access these services e.g., remoteness, financial issues, timing of clinics, services rendered.
Response: we agreed with you but some of those reasons have already been highlighted clearly in the manuscript
Query: As this study is based on data from 2018, we need to be cautious in drawing the conclusions as set out. Has the governing body taken any measures to improve health literacy? Have there been a move to build more ANC closer to the residence of clients, have there been any improvements made according the six areas of 'Safe Motherhood Initiatives'? Data based on 2018 is a limitation of this study, this needs to be stated.
Response: Agreed but some of the issues raised have not changed in the country rather ANC services are much poorer due to religious conflicts and terrorism.